# Vertical Nanoscale Vacuum Channel Triodes Based on the Material System of Vacuum Electronics

**DOI:** 10.3390/mi14020346

**Published:** 2023-01-30

**Authors:** Panyang Han, Xinghui Li, Jun Cai, Jinjun Feng

**Affiliations:** National Key Laboratory of Science and Technology on Vacuum Electronics, Beijing Vacuum Electronics Research Institute, Beijing 100015, China

**Keywords:** vertical nanoscale vacuum channel, field mission, vacuum electronics material system, environmental test, emission stability

## Abstract

Nanoscale vacuum channel triodes realize the vacuum-like transmission of electrons in the atmosphere because the transmission distance is less than the mean free path of electrons in air. This new hybrid device is the deep integration of vacuum electronics technology, micro-nano electronics technology, and optoelectronic technology. It has the advantages of both vacuum and solid-state devices and is considered to be the next generation of vacuum electronic devices. In this work, vertical nanoscale vacuum channel diodes and triodes with edge emission were fabricated using advanced micro-nano processing technology. The device materials were all based on the vacuum electronics material system. The field emission characteristics of the devices were investigated. The diode continued emitting at a bias voltage from 0 to 50 V without failure, and the current variation under different vacuum degrees was better than 2.1%. The field emission characteristics of the devices were evaluated over a wide pressure range of between 10^−7^ Pa and 10^5^ Pa, and the results could explain the vacuum-like behavior of the devices when operating in air. The current variation of the triode is better than 6.1% at V_g_ = 8 V and V_a_ = 10 V.

## 1. Introduction

Vacuum triodes led the early era of electronic devices and played important roles in amplifying, switching, or modulating electrical signals. Electrons exhibit light speed in vacuum channels due to ballistic transport. Vacuum electronic devices also have good immunity to noise and radiation. In the 1960s, development of discrete semiconductor devices rose rapidly, and they gradually occupied most of the electronic devices market due to their advantages of small size, high integration, low power consumption, low cost, etc. With the continuing development of electronic devices, vacuum electronic devices need advanced small size processing technology and new structures in order to achieve higher frequency and power, and solid-state electronic devices require smaller channel sizes to obtain higher operating speeds or even seek new materials to achieve better performance. In short, neither of them can have the comprehensive performance of size, weight, efficiency, power, stability, etc., at the same time. In fact, the field emission cathodes proposed by C A Spindt and co-workers between the 1960s to 1990s represented the first integration of vacuum electronics technology and solid-state electronics technology [1,2]. Field emission cathodes not only retain the electronic transmission characteristics of vacuum electronic devices, but also maintain the size of solid-state electronic devices. All of the above are achieved through vacuum microelectronics technology [3]. In the past three decades, intense research activity has been aimed at finding new nanomaterials for use as new cathodes for field emission devices. The nanomaterials that have received the most attention are carbon nanomaterials such as carbon nanotubes and graphene [4,5,6,7]. However, submicroscale transport distance still requires high voltage and vacuum package, which leads to the failure of device integration. In 2012, J W Han and his colleagues from the NASA Ames Research Center proposed a gate-insulated nanoscale vacuum channel transistor [8]. In the same year, H K Kim’s team from the University of Pittsburgh put forward a metal-oxide-semiconductor field-effect transistor with a vertical vacuum channel [9]. Both of the structures realized the ballistic transport of electrons in the atmospheric environment, and the operating voltages were below 10 V; thus, they were expected to be integrated into the circuit. The emergence of the nanoscale vacuum channel triode concept has drawn increasing attention [10,11]. This new hybrid device is the deep integration of vacuum electronic technology, micro-nano electronic technology and optoelectronic technology. It has the advantages of both vacuum and solid-state devices and is considered to be the next generation of vacuum electronic devices [12].

In this work, vertical nanoscale vacuum channel diodes and triodes with edge emission were prepared. The device materials were all based on the vacuum electronics material system. The field emission characteristics of the devices were investigated. The transmission channel of the device is smaller than the mean free path under atmospheric pressure, with the device showing good field emission characteristics and excellent emission stability under different vacuum degrees.

## 2. Materials and Methods

### 2.1. Materials

The reported nanoscale vacuum triodes were generally based on the Si-based semiconductor material system. Metal–air transistors with W, Pt or other metals as emitters have also been investigated [13]. In this work, we chose electrodes and insulating layers based on the material system of vacuum electronics. Molybdenum (Mo) is a metal material commonly used in vacuum electronic devices. It has good heat resistance and chemical stability. Importantly, Mo has a work function as low as 4.2 eV, which is favorable for electron tunneling and generating high current density. Due to these advantages, Mo has good field emission performance as the emission tip in the Spindt cathode. Therefore, Mo is naturally used as the field emission material in this research. For the convenience of preparation, Mo is also used as gate and anode. Finally, it is proved that Mo is compatible with advanced semiconductor manufacturing processes. Al_2_O_3_ is used as an insulating layer, and its dielectric constant is as high as 9 to 10, which is conductive to intercepting the leakage current between the cathode and the gate and improving the breakdown resistance of the device. Overall, the vacuum electronics material system is more suitable than the semiconductor material system for extreme environments such as high temperature and strong radiation.

### 2.2. Structure

The transmission distance of electrons should be close to or less than the mean free path of electrons in air at atmospheric pressure (~60 nm) [14] to avoid collision between carriers and gas molecules, so that the electronic channel can be regarded as a vacuum even if the device operates in air. Nanoscale electronic transmission channels in horizontal direction have been achieved in recent years [8,15]. A Di Bartolomeo et al. achieved side-gate graphene field-effect transistors with a self-aligned side-gate at 100 nm from the 500 nm wide graphene conductive channel. Channels and gates were formed by graphene with parallel edges. The device exhibits side-gating below 1 V with a conductance modulation of 35% and transconductance up to 0.5 mS/mm at 10 mV drain bias [16]. Keshab R Sapkota and co-workers have reported single-emitter GaN-based nanoscale vacuum electron diode devices with lateral nanogap sizes down to ~26 nm. The devices can operate in air with ultralow turn-on voltages down to ~0.24 V and stable high field emission currents (tested up to 10 µA for single-emitter devices) [17]. The lateral nanoscale channels were generally prepared by expensive exposure technology and other advanced micro/nano-fabrication technology, which makes it difficult to realize the batch production of consistent devices at the wafer level. In contrast, vertical nanoscale transmission channels can be skillfully obtained by using a multilayer stacked film structure, and the transmission distance is precisely controlled by the thickness of gate and insulating layers [9,10,18]. The vertical structure fabrication has improved operability at the wafer scale. Figure 1 shows the cross section schematic of the nanoscale vacuum triode in our research. The voltage applied between the cathode and the gate creates a strong electric field at the edge of the nanoscale cathode from which electrons are emitted due to the tunneling effect. The electrons show ballistic transport in the vacuum channel and, finally, are absorbed by the anode. Each electrode is patterned with a pad for electrical test.

### 2.3. Fabrication

In order to achieve practical applications and truly become the next generation of electronic devices, nanoscale vacuum triodes need to be integrated into the circuit to operate. Therefore, advanced semiconductor manufacturing technology was adopted to realize the batch manufacturing of nanoscale vacuum triodes on the wafer. The whole process was completed on a 3 inch n-Si (100) substrate, with 0.8 µm SiO_2_ being formed on the wafer by thermal oxidation technology to isolate the device from the substrate. The semiconductor-free devices are unaffected by the substrate during operation. An amount of 15 nm Mo was deposited on the substrate by electron beam evaporation and patterned by ultraviolet lithography as a cathode, after which 20 nm Al_2_O_3_ and 15 nm Mo were deposited successively, and Mo was patterned as a gate. The second step was repeated to obtain an anode. Finally, a 1 × 1 × 1 μm^3^ square well was etched in the middle of the overlapping electrode from the top anode to the Si substrate by focused ion beam etching (Figure 2), which was the vertical vacuum channel. The emission area was determined by the perimeter. The length of the channel was 1 µm, while the transmission distance of electrons emitted from cathode edge to anode was 55 nm (less than 60 nm), which was determined precisely by the thicknesses of insulating layers and gate (Figure 1). Figure 2 shows the active part of the device. The sizes of cathode, gate, and anode from bottom to top are 8 µm, 6 µm and 4 µm, respectively. The illustration in the upper left corner shows the multilayer film structure with clear interfaces.

### 2.4. Characterization

The devices were measured in a customized vacuum test system. The dry pump renders the minimum pressure of the vacuum chamber as low as 10^−7^ Pa. The pressure in the vacuum chamber is gradually increased by adjusting the opening of the gate valve. The nitrogen pipeline connected to the vacuum chamber provides clean nitrogen. The external power supply continuously adjusts the voltage, and the accuracy of the voltage is 0.01 V. The gate current and anode current are displayed on the Keithley 6485 picoammeter. All tests were conducted at room temperature.

## 3. Results

Although the vertical transmission distance is designed to be smaller than the mean free path, which means the device could work in air-ambient conditions, the initial measurement was still conducted under a vacuum of about 10^−7^ Pa to ensure a clean emission surface and a stable test status. A diode was first prepared based on the fabrication method in Subchapter 2.3. to verify emission characteristics. The transmission distance between the two electrodes was 55 nm. A 1 × 1 × 1 µm^3^ square well was etched into the Si substrate in the middle of the upper anode. The anode current versus anode voltage curve is shown in Figure 3. The turn-on voltage (defined as the voltage needed to generate 1 nA current) is 0.5 V. The inset is plotted based on the Fowler–Nordheim (F–N) equation:$$\frac{I}{V^{2}}=ae^{-b/V}$$
where *a* and *b* are constant. The curve exhibits obvious F–N behavior in the high electric field (low 1/V_a_), which proves that electrons are emitted from the cathode through FN tunneling. In the low electric field (high 1/V_a_), the slope of the curve changes. That may result from other mechanisms, e.g., the space-charge-limited current mechanism [18].

The cross section of the well exhibits that the upper electrode and the lower electrode are symmetrical, and they have identical materials and effective working surfaces. If the lower electrode is grounded, the upper electrode is used as the anode. On the contrary, if the upper electrode is grounded, the lower electrode is used as the anode. The reverse voltages were applied to the electrodes for measurement (Figure 4). Since the upper electrode was close to the opening of the well, it could only collect a fraction of electrons emitted from the cathode when it was used as the anode (red line). While the lower electrode was in a closed space formed by Si/SiO_2_, it could collect more electrons (black line). When the applied voltage was 15 V, I_black_ is 21% higher than I_red_.

The designed device shows excellent stability of operation, which is one of the important conditions for application. Figure 5a shows the current–voltage plot of the device with a transmission distance of 55 nm. The current increases rapidly with the voltage changes between 25 to 32 V, which shows a typical strong field emission phenomenon. The electrons emitted from the cathode are accelerated by the high electric field derived from the applied voltages. When the applied voltage is 30 V, the emission current is 1.79 µA, and the corresponding emission current density is about 3000 A/cm^2^. For V_a_ > 32 V, the growth of emission current tends to be slow. The device can continue emitting before V_a_ reaches 50 V without damage and failure. Due to the test system configuration, the performance was not tested over 50 V. The performance comes from the emission of electrons from the edges, whose stability is better than that of the tip emitters.

Vacuum nanoscale electronic devices have been expected to operate in air, so the field emission characteristics of the designed devices under different vacuum degrees (a wide pressure range between 10^−7^ Pa to 10^5^ Pa) were evaluated (Figure 5b). The device was installed in a customized closed chamber, which was pumped down to a base pressure of 10^−7^ Pa by dry pump. Controlling the opening of the gate valve increased the pressure gradually. Clean nitrogen provided pressure over 10 Pa to 10^5^ Pa. I−V curves under each vacuum degree were measured several times and there were no obvious differences. The field emission characteristic curves between 10^−6^ to 10^−2^ Pa show a dependence on vacuum degrees. With the decrease of vacuum degree, the emission current decreases as a whole. This may be related to the collision between emitted electrons and gas molecules. However, the curves under other vacuum degrees did not conform to the law. The main reason for this result may be the surface state of the cathode. That is, the adsorption state of the gas molecules changed the work function of the emission surfaces, thereby changing the emission state. Nevertheless, different vacuum degrees will generally not seriously affect the working performance of the device, and the emission currents are in the same order of magnitude, which can explain the vacuum-like behavior of the devices when they operate in air. Given a fixed applied voltage of 10 V, the stability of the emission current was studied [19]. The current variation with the varying vacuum degree was better than 2.1%.

The triode performances of the nanoscale vacuum device were also tested. Figure 6a shows the anode current versus anode voltage characteristics with a fixed gate bias voltage of 10 V, 15 V, and 25 V. The curve with the maximum gate bias voltage (blue curve) has the minimum turn-on voltage. In a triode the gate should theoretically play a role in modulation. However, the modulation by the gate seems rather limited in Figure 6a. The reasons are analyzed as follows. Initially, the cathode-to-gate distance is close to the gate-to-anode distance, and all electrodes are metal, which could in theory emit electrons at an adequate voltage bias. When V_g_ and V_a_ are applied and the V_a_ > V_g_, electrons are emitted not only from the cathode, but also from the gate. In addition, when V_g_ > V_a_ at the beginning of the test, electrons may be emitted from the cathode and anode simultaneously and collected at the gate. Last but not least, because the gate is located in the middle of the three electrodes, it will intercept the electrons transferred from the cathode to the anode [20]. In short, the structure and metal electrodes cause the crosstalk of electrons, thus weakening the modulation of the gate. Several studies have been carried out to improve the modulation. I J Park et al. proposed a new slit-type vacuum channel transistor. The gate-to-anode distance is 30 nm, while the cathode-to-gate distance is only 2 nm. The ratio of the cathode-to-gate distance to the channel length is smaller than that of other vertical vacuum channel triodes. Consequently, the device exhibits better control on electron emission and can operate without gate leakage when V_a_ and V_g_ are the same [10]. Analogical work conducted by W T Chang and co-workers also clarifies that the distance between cathode and gate should be as thin as possible to obtain a high-drive current [18]. The reduction of the cathode-to-gate distance will increase the parasitic capacitance and also lead to the tunneling leakage current. For this issue, J W Han et al. put forward a device with an extended gate structure; that is, the gate is folded near the cathode edge region. The gate oxide is thick in the field region, and conversely, it is thin near the cathode edge. The structure maximizes the cathode-to-gate controllability while minimizing the parasitic capacitance formed by the cathode fanout field region [21]. The emission stability of nanoscale vacuum triode under different pressures is shown in Figure 6b. The test was conducted at V_g_ = 8 V and V_a_ = 10 V. The maximum of I_a_ is 1173 nA and the minimum of I_a_ is 1102 nA. This suggests that the current variation is better than 6.1%. Although most of the electrons are collected by the anode, a gate current greater than 100 nA is unexpected. It may be that the symmetrical structure and the short transmission distance cause the electronic crosstalk between electrodes. The specific mechanism needs further study.

## 4. Conclusions

In this paper, vertical nanoscale vacuum channel diodes and triodes with edge emission were fabricated using advanced micro-nano processing technology. The field emission characteristics of the devices were investigated. Good F−N behavior in the high electric field was obtained, which proves that electrons are emitted from the cathode through FN tunneling. The test results of the devices under a wide pressure range between 10^−7^ Pa to 10^5^ Pa explained the vacuum-like behavior of the devices when operated in air. The devices exhibited excellent voltage resistance and emission stability without failure. All these characteristics derive from the special structure and material design of the devices. Firstly, the transmission distance of electrons is less than the mean free path; secondly, the device materials are all based on the vacuum electronics material system; thirdly, electrons are emitted from edge emitters rather than tip emitters. All of the above provide conditions similar to operation in a vacuum, allowing high field emission stability even in different pressure environments. The designed hybrid devices are the deep integration of vacuum electronics technology and micro-nano electronics technology, and they have the advantages of both vacuum and solid-state devices. Therefore, they are expected to realize large-scale vacuum-like integrated circuits, which could operate in severe high temperature and radiation environments [22]. Further environmental experiments are worth exploring.

## Figures and Tables

**Figure 1 micromachines-14-00346-f001:**
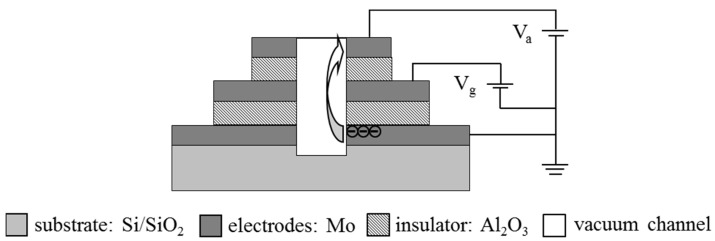
Schematic of the vertical vacuum triodes.

**Figure 2 micromachines-14-00346-f002:**
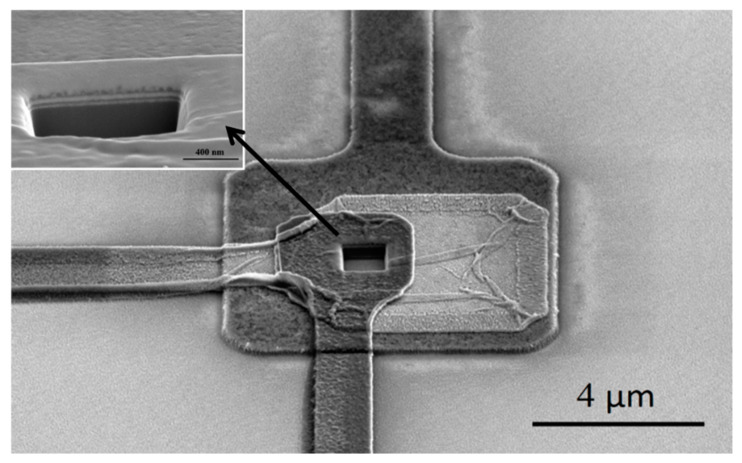
Scanning electron micrograph of active part of the device. The illustration shows the multilayer film structure (at 20° tilt to horizontal), which is consistent with Figure 1.

**Figure 3 micromachines-14-00346-f003:**
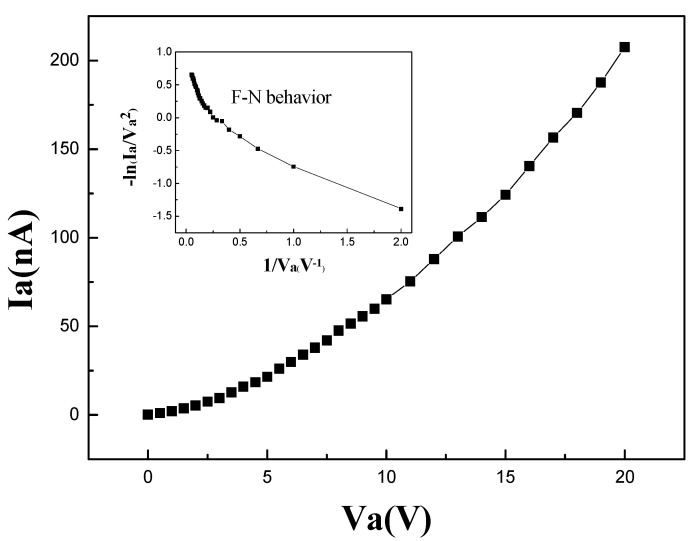
The anode current versus anode voltage curve of a diode at a bias voltage from 0 to 20 V. Inset: F–N plot for the I–V curve.

**Figure 4 micromachines-14-00346-f004:**
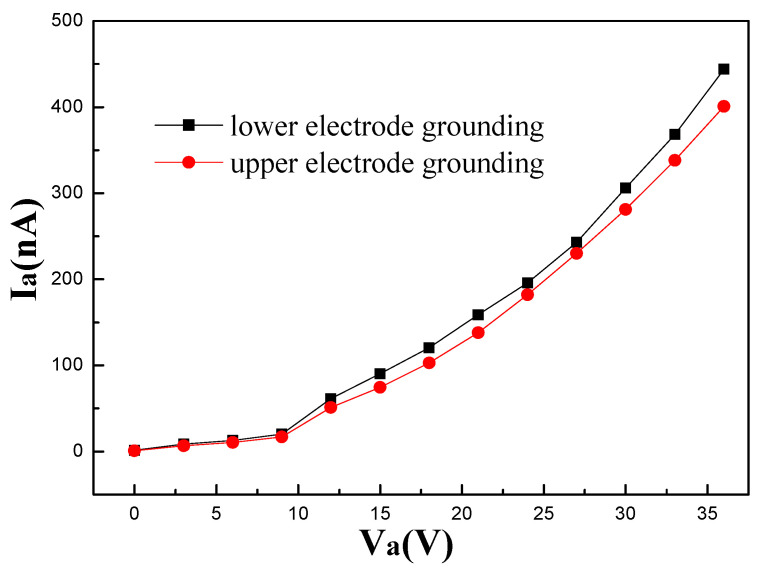
Current–voltage curves of lower electrode and upper electrode grounding.

**Figure 5 micromachines-14-00346-f005:**
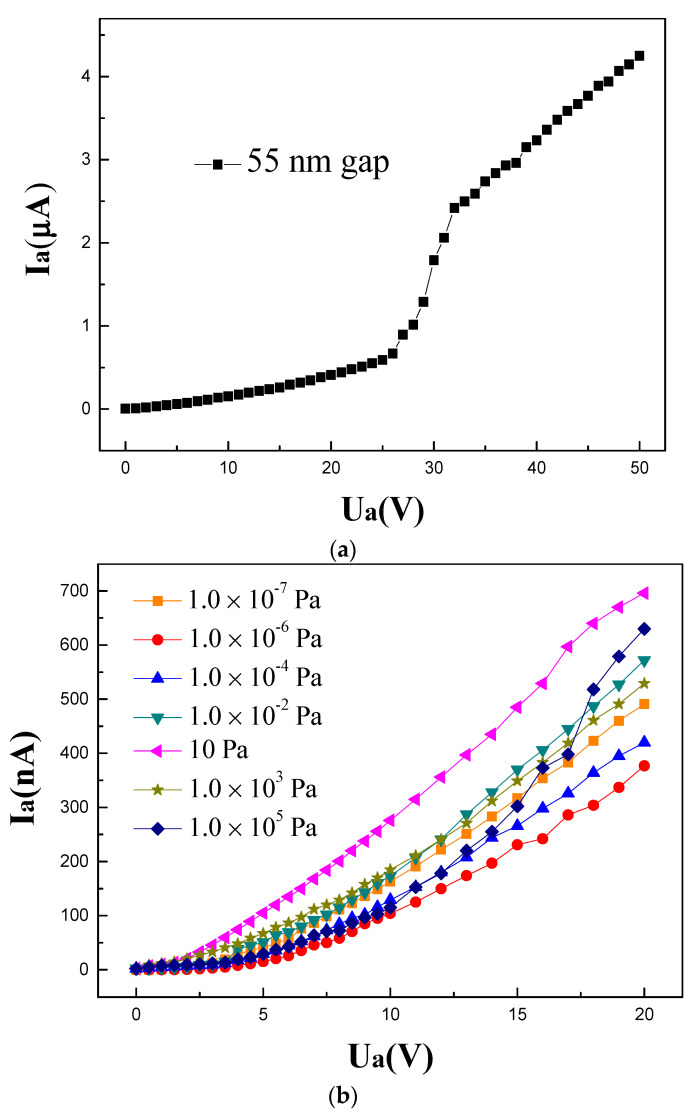
(**a**) Current–voltage plot of the device with a transmission distance of 55 nm. (**b**) Field emission characteristics under different vacuum degrees.

**Figure 6 micromachines-14-00346-f006:**
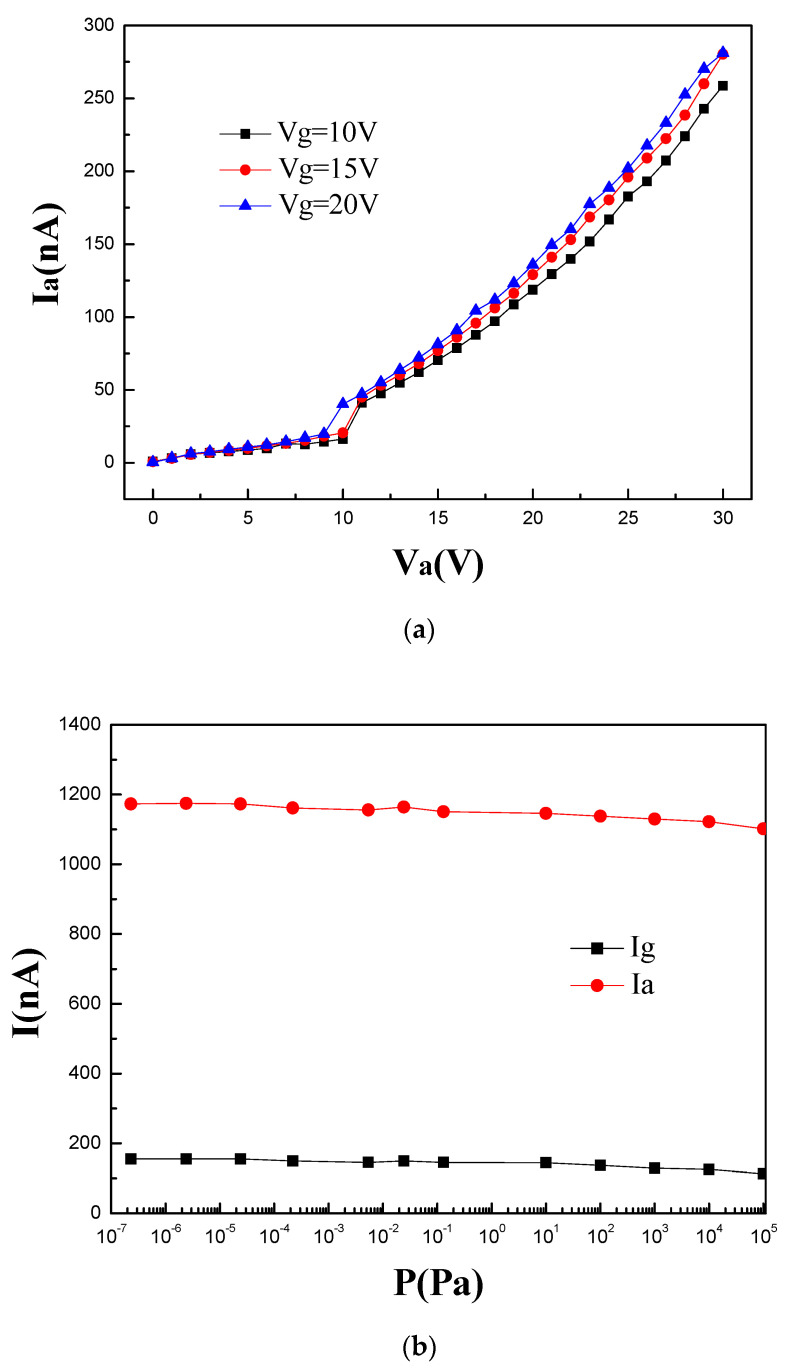
(**a**) Anode current versus anode voltage characteristics of a triode at V_g_ = 10 V, 15 V, or 25 V. (**b**) Emission stability with different pressures at V_g_ = 8 V and V_a_ = 10 V.

## Data Availability

Not applicable.

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
