# Peer review of "Vertical Nanoscale Vacuum Channel Triodes Based on the Material System of Vacuum Electronics"

_micromachines, 2023, doi:10.3390/mi14020346_

Round 1
Reviewer 1 Report
The authors present a vertical nanoscale vacuum channel triode based on material system of vacuum electronics. The field emission characteristics under different vacuum degrees was investigated. The results are interesting and of value. Therefore, I am supportive of publication after the revision of the following comments:
1. The authors should clarify the definition of emission stability. How do the authors get the values of the emission stability (2.1%, 6.1%)?
2. Why there is a jump of the drain current when the voltage is between 25V and 32V in line 160? The authors should add some explanations about this.
3. The authors need to make clear that the horizontal coordinate of Figure 4 is Va or Vg?
4. Since the gate is closer to cathode than anode, according to FN tunneling mechanism, the influence of gate voltage on channel current should be stronger than that of anode voltage. But it seems that the influence of gate voltage on channel current is relatively weaker in Fig.6(a). Can the authors explain it?
5. Since the stability is an important indicator of this device, deviation analysis of repeated measurements of the same device is also important. Can the authors add some measurement results on it?
6. The manuscript is missing multiple significant recent references in the field of vacuum/air channel devices or transistors.
7. Subchapter number in line 94 and 118 should be revised.
8. Fig.5 (a) and (b) are misplaced.
I will also give a friendly hint according to the author’s confusion about the gate leakage current in Fig.6 (b). Since the quasi-two-dimensional electron gas mentioned in Ref. 4 will also accumulate in gate, the edge emission will not only occur on sidewall of the defined cathode but also the gate.
Reviewer 2 Report
In this paper, the authors present a vertical nanoscale vacuum channel diode-triode with edge emission, fabricated over a Si/SiO2 substrate with Mo and Al2O3 films. The field emission characteristics of the device are studied under a wide pressure range.
The presented device is not new but the study is interesting and offers useful data. The paper could be reconsidered for publication in the journal after a major revision.
Here are points that the authors should consider:
1. There are typos to correct. Make careful proofreading. Examples: “ hige voltage”, “2.2. sturcture”, “the gate palys”, etc.
2. “The field emission cathode not only retains the electronic transmission characteristics of vacuum electronic devices but also has the size of solid-state electronic devices. All above are realized by vacuum microelectronics technology.” The authors could mention here that in the past three decades there has been intense research activity aiming at finding new nanomaterials to use as new cathodes for field emission devices. The most considered nanomaterials have been carbon nanomaterials such as carbon nanotubes and graphene (see for instance https://doi.org/10.1088/1361-6528/aa96e6)
3. “It is very difficult to achieve nanoscale electrode spacing, that is, electronic transmission channel in horizontal direction, which requires advanced and expensive exposure technology.” I agree with this statement. However, the authors could note that attempts in making horizontal graphene field emission devices have been done (see for instance https://doi.org/10.1063/1.4958618, which could be added to the citation list).
4. “The transmission distance of electrons should be close to or less than the mean free path in atmospheric pressure (~60 nm) to avoid the collision between carriers and gas molecules,… “ How is the 60 nm mean free path estimated?
5. Increase the font size of the numbers in the inset of figure 3, which are not readable.
6. Specify the gate voltage for the data reported in figures 3, 4, and 5.
7. Figure 5: (a) and (b) captions are swapped.
8. “The current is still jumping with the voltage 160 changes between 25 to 32V.” Is there any explanation for this jump?
9. About figure 6: “The turn-on voltages of the characteristic curves decrease with the 192 increase of the gate bias voltages.” This is not very clear from the plot. Perhaps plotting the current on a log scale would help. Besides, how is the turn-on field defined?
10. “At the same anode voltage, large gate bias voltage 193 brings large anode current. To sum up, the gate palys an important role in modulation.” I disagree with this conclusion. The modulation by the gate seems rather limited. The authors should revise or substantiate this statement.
Round 2
Reviewer 1 Report
The manuscript has been carefully revised and can be published in the present form.
Reviewer 2 Report
I appreciate the attention given to all my comments and suggestions. The authors made changes and improvements in their manuscript and also corrected some technical errors. They gave convincing responses to the various questions and comments that I had raised.
The revised version of the manuscript appears complete and technically sounder. The paper can be accepted for publication in its current form.